# Rift Valley Fever Virus Non-Structural Protein S Is Associated with Nuclear Translocation of Active Caspase-3 and Inclusion Body Formation

**DOI:** 10.3390/v14112487

**Published:** 2022-11-10

**Authors:** Lukas Mathias Michaely, Melanie Rissmann, Federico Armando, Felicitas von Arnim, Markus Keller, Martin Eiden, Rebecca König, Benjamin Gutjahr, Wolfgang Baumgärtner, Martin H. Groschup, Reiner Ulrich

**Affiliations:** 1Center of Systems Neuroscience, Department of Pathology, University of Veterinary Medicine Hannover, Foundation, 30559 Hannover, Germany; 2Institute of Novel and Emerging Diseases, Friedrich-Loeffler-Institute, Insel Riems, 17493 Greifswald, Germany; 3Partner Site Hamburg-Lübeck-Borstel-Riems, German Center for Infection Research (DZIF), Insel Riems, 17493 Greifswald, Germany; 4Erasmus Medical Center, Department of Viroscience, 3015 GD Rotterdam, The Netherlands; 5Department of Experimental Animal Facilities and Biorisk Management, Friedrich-Loeffler-Institute, Insel Riems, 17493 Greifswald, Germany; 6Institute of Veterinary Pathology, Faculty of Veterinary Medicine, Leipzig University, 04103 Leipzig, Germany

**Keywords:** caspase-3, Rift Valley fever, apoptosis, inclusion bodies

## Abstract

Rift Valley fever phlebovirus (RVFV) causes Rift Valley fever (RVF), an emerging zoonotic disease that causes abortion storms and high mortality rates in young ruminants as well as severe or even lethal complications in a subset of human patients. This study investigates the pathomechanism of intranuclear inclusion body formation in severe RVF in a mouse model. Liver samples from immunocompetent mice infected with virulent RVFV 35/74, and immunodeficient knockout mice that lack interferon type I receptor expression and were infected with attenuated RVFV MP12 were compared to livers from uninfected controls using histopathology and immunohistochemistry for RVFV nucleoprotein, non-structural protein S (NSs) and pro-apoptotic active caspase-3. Histopathology of the livers showed virus-induced, severe hepatic necrosis in both mouse strains. However, immunohistochemistry and immunofluorescence revealed eosinophilic, comma-shaped, intranuclear inclusions and an intranuclear (co-)localization of RVFV NSs and active caspase-3 only in 35/74-infected immunocompetent mice, but not in MP12-infected immunodeficient mice. These results suggest that intranuclear accumulation of RVFV 35/74 NSs is involved in nuclear translocation of active caspase-3, and that nuclear NSs and active caspase-3 are involved in the formation of the light microscopically visible inclusion bodies.

## 1. Introduction

Rift Valley fever phlebovirus (RVFV, Phenuiviridae) causes Rift Valley fever (RVF), a zoonotic and potentially lethal disease that is currently endemic in Africa and on the Arabian Peninsula [1,2]. Although approximately 50% of infected humans experience no or only mild signs of the disease, the infection can lead to severe or even lethal disease and its complications include renal failure, hemorrhagic fever, hepatitis, encephalitis, ocular complications and possible life-long impairment or even death [2,3,4]. While the disease can pose a threat to public health systems, the impact of Rift Valley fever outbreaks on agricultural economic welfare is of equal or even greater concern [5,6]. Within ruminant livestock, the disease leads to trimester indiscriminate “abortion storms” [7]. Especially among the most susceptible species of sheep, these outbreaks cause high mortality rates among neonatal offspring, thus eliminating complete generations of local ruminant herds [7,8,9]. Therefore, RVF has been recognized as an important emerging disease that warrants further investigation due to limited understanding of its pathogenesis as well as its possible spread towards non-endemic areas [3,10,11].

To address this lack of knowledge and to investigate RVF pathogenesis, a panel of susceptible animal models has been established including ruminants, rodents and monkeys [12,13,14,15,16,17]. It is known that RVF pathogenesis and virulence differs among infected species and even between strains within the same animal model [3]. As described by Kwaśnik et al. (2021), the course of disease can broadly be divided in three categories: (i) cases of severe and acute illness are associated with an uncontrolled viremia marked by high viral loads. (ii) The second pattern exhibits quickly decreasing levels of viremia and mild to no clinical disease. (iii) Third, fever and viremia are displayed during the initial onset of disease, but the main hallmark is the development of delayed signs such as encephalitis or ocular disease, which in humans can manifest weeks to months after the initial infection and the pathogenesis of which remains unclear so far [2,3,10].

On the cellular level, the virus exhibits a high tropism towards mononuclear phagocytes and hepatocytes thus inducing hepatic disease among other signs [2,3]. The virus enters hepatocytes by endocytosis after its viral glycoproteins interact with yet to be unequivocally identified receptors [3,18]. Possible cellular surface receptors include liver/lymph node-specific intracellular adhesion molecules-3 grabbing non-integrin (L-SIGN) cellular C-type lectins that resemble the dendritic cell-specific intercellular adhesion molecule-3-grabbing non-integrin (DC-SIGN)-mediated viral entry into dendritic cells [3,18,19]. Thereafter, the innate immune response may control the viral spread by several mechanisms including a strong interferon (IFN) type I response as seen in many virus infections [2,18,20]. RVFV has developed evasive strategies to counteract this innate immune signaling, mainly due to interactions of the multifunctioning non-structural protein S (NSs) [2,6,18,20]. Another important part of a successful antiviral immune response is represented by quick apoptosis of infected cells via the activation of the Janus kinase and signal transducer and activator of transcription protein (JAK-STAT) pathway [21,22]. This pathway and the follow-on process of cellular apoptosis includes a group of proteases, called caspases, a designation derived as an acronym of their function as cysteinyl-aspartate-specific protease [23]. Caspases are one of the key components during apoptosis with initiating (caspase 8 or caspase 9) as well as effector functions (caspase-3, caspase 6, caspase 7) and have thus been used as an easily accessible marker of apoptosis in several studies [24,25,26,27,28].

While these molecular mechanisms are typically not accessible by light microscopy, an occasional feature of RVF-induced lesions is the occurrence of intranuclear, eosinophilic inclusions [7,22,29,30]. These inclusions have been subject to different investigations, and it has been shown that nuclei of infected cells contain filamentous structures of viral NSs protein [31,32,33,34]. Positive staining results of affected hepatocytes via terminal desoxynucleotidyl transferase dUTP nick end labeling (TUNEL) assay indicate that RVFV replication induces hepatocellular apoptosis [2,29]. However, the role of caspase-3, its interaction with NSs and intranuclear inclusion body formation has yet to be elucidated.

The present study uses archival tissue samples from two mouse strains that have been used in a previously reported infection experiment [35]. C57Bl/6-IFNAR^tmAgt^ mice (IFNAR^−/−^) are a strain of knockout mice that lack the expression of the interferon type A receptor (IFNAR) and therefore are unable to mount a successful interferon response [36,37]. As described above, this immunodeficiency is required to induce severe hepatitis with the attenuated RVFV strain MP12 [36,37]. The second mouse strain of heterozygous Crl:NU(NCr)-Foxn1^nu^ mice (hzNU), infected with RVFV 35/74 was used previously as a control group for homozygous NU mice and carries the NU mutation [35]. Homozygous NU (nude) mice lack thymic development and therefore are severely compromised in their cellular immune response [35,38]. The heterozygous mice used in this study however do not develop an immunodeficient phenotype and can be considered wild-type mice for this study’s purposes [35]. The hypothesis of this study is that a strong expression of NSs induced by the virulent 35/74 strain in immunocompetent mice but not the attenuated MP12 strain in immunodeficient mice leads to intranuclear co-localization of both RVFV NSs and active caspase-3. Therefore, the combined accumulation of NSs and activated caspase-3 within the nucleus is suggested to be the mechanism leading to the formation of eosinophilic, comma-shaped, intranuclear inclusion bodies described in severe cases of RVF. Based on this premise, the aims of the present study are the detection and characterization of intranuclear caspase-3 and NSs accumulations, as well as intranuclear inclusion bodies in liver lesions induced by two RVFV strains with either functional or defective NSs, respectively, in mice.

## 2. Materials and Methods

### 2.1. Virus

RVFV strain 35/74 (accession number: JF784386-88) was propagated in BHK-21 cells (Collection of Cell Lines in Veterinary Medicine, Friedrich-Loeffler-Institut, Germany) using minimum essential medium (MEM) supplemented with 2% FBS at a temperature of 37 °C and 5% CO_2_. RVFV strain MP12 was propagated on Vero-76 cells, following previously published protocols [35,39,40]. The supernatant of both propagations was harvested, at 80% cytopathic effect (CPE) and the TCID_50_ assay was used to determine the virus titer according to Spearman and Kaerber [41].

### 2.2. Mouse Experiments and Sampling

All tissues used in the current study were archival specimens from a previously published study concerning the importance of a functional innate and adaptive immune response to overcome RVFV infection, which comprised nine immunodeficient mouse strains as well as four background strains [35]. Formaldehyde-fixed, paraffin-embedded (FFPE) liver samples from five to seven-week-old, female, infected mice that had to be euthanized due to severe signs of disease and exhibited hepatic lesions were selected from the archives along with mock-infected control animals of the same strain and age (Table 1). All of the experiments were approved and authorized by the local authority (Landesamt für Landwirtschaft, Lebensmittelsicherheit und Fischerei Mecklenburg-Vorpommern (permission 7221.3-1-038/17 (19 September 2017)) and performed in accordance with the German regulations and legal requirements.

Immunodeficient C57Bl/6-IFNAR^tmAgt^ mice (IFNAR^−/−^) were bred at the Friedrich-Loeffler-Institute mouse stock, while immunocompetent, heterozygous Crl:NU(NCr)-Foxn1^nu^ mice (hzNU) were commercially obtained from Charles River Laboratories (Sulzfeld, Germany) [38]. Groups of three mice were kept in individually ventilated cages (IsoCage N, Tecniplast, Hohenpeißenberg, Germany). Mice received water and standard mouse food (ssniff Spezialdiäten GmbH, Soest, Germany) ad libitum. Infection of the mice was performed by subcutaneous injection of 0.1 mL virus suspension with a TCID_50_ of 1.4 × 10^3^ per animal of RVFV 35/74 or a TCID of 1.43 × 10^3^ of RVFV MP12 per animal, respectively. Mock-infected control mice received virus-free cell culture medium. Mice were euthanized at a predefined severity score by isoflurane anesthesia and subsequent cardiac exsanguination. hzNU mice represent the background strain for severely immunocompromised NUDE mice that were used in the aforementioned, original study, and do not show any phenotype of the NU mutation [35,38]. Therefore, they can be considered the wild-type for the current study. The inclusion of the immunodeficient IFNAR^−/−^ mice as a second mouse strain is required due to their unique susceptibility towards RVFV MP12 [35,36,37].

### 2.3. Histology and Immunohistochemistry

Liver samples were fixed in 4% neutral buffered formaldehyde for 21 days and embedded in paraffin wax. Two µm sections were cut and used for immunohistochemistry (IHC) and hematoxylin-eosin (HE) staining. IHC for RVFV proteins and active caspase-3 detection was performed according to previously published protocols [15,39,42]. Briefly, it included an optional antigen retrieval, blocking of nonspecific background signals via the application of serum from the respective host species, followed by overnight incubation of the primary antibodies, application of biotinylated secondary antibodies (biotinylated goat anti rabbit, BA-1000, or biotinylated rabbit anti sheep, BA-6000, Vector Laboratories, Burlingame, CA, USA), application of the avidin–biotin–peroxidase complex method (ABC, Elite PK6100; Vector Laboratories, Burlingame, CA, USA) using 3-amino-9-ethylcarbazol (AEC, Dako, Glostrup, Denmark) as a chromogen and hematoxylin counterstain [15,39,42]. Primary antibodies and protocol details are shown in Table 2. Monoclonal anti-RVFV antibodies were raised against recombinant bacterial proteins and have been subject to a comprehensive examination [34,39,42]. RVFV-infected and non-infected Vero cells were processed as FFPE blocks and used as positive or negative controls, respectively.

Histopathological examination and cell count per mm^2^ was carried out manually, using a light microscope (Carl Zeiss, Oberkochen, Germany) with a Zeiss Kpl-W10x/18 ocular and a Zeiss 40/0.65 objective. Five representative high-power fields (HPF, 1 HPF equaling 0.159 mm^2^) were counted per sample. An Olympus BX51 microscope and a DP72 Camera with the manufacturer’s operating software cellSens, version 1.18 (Olympus solution, Tokyo, Japan) was used for digital photography.

### 2.4. Immunofluorescence

FFPE mouse liver tissue from a RVFV 35/74-infected hzNU mouse and a mock-infected control mouse was used for immunofluorescence to verify the co-localization of IHC expression of RVFV NSs and active caspase-3. Immunofluorescence double staining was carried out according to previously published protocols [43,44]. Briefly, tissue sections were pretreated for antigen retrieval using citrate buffer (pH 6). Following the blocking of nonspecific bindings with goat serum, sections were incubated with primary antibodies (Table 2) targeting RVFV NSs and active caspase-3 at 4 °C overnight. Thereafter, sections were incubated with secondary cyanine 2-conjugated goat anti mouse (Dianova, Hamburg, Germany) and cyanine 3-conjugated goat anti rabbit (Invitrogen, Carlsbad, CA, USA) antibodies, respectively, for 60 min at room temperature. Nuclear counterstain and mounting were performed using 0.01% bisbenzimide (diluted in Aqua bidestillata; bisbenzimide, Sigma-Aldrich Chemie GmbH, Taufkirchen, Germany) for 10 min and mounted using Fluoroshield mounting medium (Sigma-Aldrich, St. Louis, MO, USA) and coverslips. Fluorescence pictures were taken using a fluorescence microscope (BZ-9000E, Keyence Deutschland GmbH, Neu-Isenburg, Germany).

### 2.5. Statistical Analysis

Statistical workup of the cell count data was performed via the Mann–Whitney U-test using Graphpad Prism (GraphPad Software, Inc., San Diego, CA, USA, version 9.0.0) and SAS 9.4 (SAS Institute, Inc. Cary, NC, USA).

## 3. Results

### 3.1. Clinical Signs

After infection, the mice either died or had to be euthanized due to severe signs of disease that included ruffled fur, lack of unprovoked and provoked movement, tachypnea and apathy at 2 or 3 days post infection (dpi).

### 3.2. Histology and Immunohistochemistry

The liver samples of all the RVFV-infected mice exhibited severe diffuse hepatic necrosis, with it affecting over 90% of the analyzed organ tissue (Figure 1). Likewise, numerous condensed, rounded, eosinophilic hepatocytes with pyknotic and karyorrhectic nuclei, suggestive of apoptotic hepatocytes were present, while leukocytic infiltrates were mostly absent. IHC labeling revealed abundant and diffuse extra- and intracellular RVFV nucleoprotein (Np) expression (Figure 1).

Nuclear, comma-shaped, eosinophilic inclusions, mostly associated with complete margination of chromatin, were present in intralesional hepatocytes within RVFV 35/74-infected hzNU mice but were absent in RVFV MP12-infected IFNAR^−/−^ mice. Likewise, in RVFV 35/74-infected hzNU mice, RVFV NSs and active caspase-3 immunoreactivity was present within hepatocytes with a comparable, intranuclear, comma-shaped accumulation of both signals. No intranuclear expression of RVFV NSs or active caspase-3 was detected in RVFV MP12-infected IFNAR^−/−^ mice (Figure 1).

The control mice did not exhibit hepatocellular necrosis, apoptosis or inflammatory lesions. No immunolabeling for RVFV proteins or active caspase-3 was observed (Table 3).

### 3.3. Immunofluorescence

Liver tissue from a RVFV 37/74-infected hzNU mouse was double labeled for proof of principle analysis of nuclear co-localization of NSs and active caspase-3. Immunofluorescence showed intranuclear co-expression of RVFV NSs and active caspase-3 (Figure 2). HE-stained serial slides from the same FFPE blocks revealed intranuclear, comma-shaped, eosinophilic inclusions with a comparable morphology. (Figure 2). Negative controls from a mock-infected mouse did not show any immunolabeling.

## 4. Discussion

The initial observation of severe, diffuse, hepatic necrosis with virus strain-dependent presence or absence of inclusion bodies was made during the histopathological analysis of an animal experiment designed for a previously published study [35]. The specimens were re-used for the current manuscript. The strains were selected under the premise to compare two groups in which either attenuated RVFV MP12 or the virulent RVFV 35/74 induces hepatic necrosis of comparable severity [35,40]. The hypothesis is that intranuclear inclusion bodies are formed by nuclear co-localization of NSs protein and active caspase-3 only in RVFV NSs-infected hzNU but not in RVFV MP12-infected IFNAR^−/−^ mice.

Considering that only two conditions, differing in both virus strain and host, are reported here, the available choice of archival samples is a limitation of this study and it should be admitted that a complete experimental design would include RVFV 35/74-infected IFNAR^−/−^ mice and RVFV MP12-infected hzNU mice for unequivocal proof that the observed effect is solely due to the different virus strains. Specimen of RVFV 35/74-infected IFNAR^−/−^ mice are currently unavailable, since they were not needed for the hypothesis of the initial study [35]. However, the lacking results can be anticipated with a high degree of certainty, since it is well known that 35/74 infection leads to severe hepatic necrosis with intranuclear inclusions in other immunocompetent mouse strains [40]. Furthermore, our previous study has shown that MP12 does not induce clinical disease and no hepatic lesions were observed at 14 dpi in the immunocompetent hzNU mice [35]. In all RVFV-infected animals, RVFV Np was expressed diffusely within hepatic lesions, however, intranuclear expression of NSs and pro-apoptotic active caspase-3 was present only in RVFV 35/74-infected mice. In agreement with the results, virulent strains of RVFV are known to induce hepatocellular apoptosis and intranuclear eosinophilic inclusions in mice and lambs [3,7,40,45]. Using a limited number of sheep experimentally infected with RVFV 35/74, comparable intranuclear inclusion bodies, as well as intranuclear NSs and active caspase-3 accumulation, were observed in lambs with severe, coalescing, necrotizing hepatitis, but not in adult sheep with mild to moderate, multifocal, necrotizing hepatitis (unpublished observations).

The NSs protein consists of 265 amino acids, is encoded on the small segment of the viral genome and is one of the main pathogenicity factors of RVFV [18,46,47]. As in the related viruses, Toscana virus, Bhanja virus, and Uukuniemi virus, the protein accumulates within the cytoplasm during viral replication, but differs from other structural and non-structural viral proteins by also forming filamentous structures within the nucleus [33,47,48,49]. It is unclear, whether this trait is unique to RVFV as the localization of NSs in other phlebovirus infections is unknown so far [49]. The intranuclear filaments of RVFV NSs are formed by the C-terminal amino acids of NSs and, as described in the current study, these NSs-filaments are most likely involved in the formation of the eosinophilic, comma-shaped, intranuclear inclusions occasionally observed by light microscopy in RVF hepatitis [22,30,33,47].

NSs plays a significant role in the evasion from host innate immunity by inhibiting IRF-3, NF-κB and AP-1, which are transcription factors inducing IFN-β expression [47,50]. Furthermore, NSs binds to the subunit protein p44 of TFIIH, which is an important transcription factor for RNA polymerase I and II [47,51]. NSs also promotes post-translational degradation of dsRNA-dependent PKR and arrests cell cycle progression by hitherto unknown mechanisms [47,52,53,54]. Additionally, NSs can activate the classical DNA damage-signaling pathway via the ATM kinase, inducing phosphorylation of p53, and activation of Chk.2 and H2A.X [47,55,56,57]. Among 33 NSs-interacting genes investigated in a large chromatin immunoprecipitation study, downregulation of the genes responsible for protein phosphorylation, ubiquitination, immune responses, apoptosis, transcription or development was observed [47,57]. However, this study did not include caspase-3, but only the related caspases 8 and 9, the latter of which was significantly downregulated by NSs activity [57]. Furthermore, NSs facilitates viral translation by sequestration of poly(A)-binding protein 1 and downregulation of protein kinase PKR [52,53,58]. Finally, the involvement of NSs in coagulative disorders has been suggested, which is one of the major hallmarks of human hemorrhagic fever in severe cases of RVF [48,57]. Due to this plethora of adverse effects, NSs has been recognized as a major pathogenicity factor and has been targeted by vaccine developers and researchers [47].

Innate mechanisms such as apoptotic pathways represent a very important first line of defense to contain early virus replication and spread [2,3,20,29]. The effect of RVFV on apoptotic pathways is not fully understood. It is known that intranuclear presence of NSs induces apoptosis by its inhibitory effect on the cell cycle and subsequent activation of p53 [22,59]. On the other hand, there is evidence that NSs inhibits apoptosis [59], and RVFV non-structural protein NSm has also been shown to have an apoptosis-blocking effect [60]. In the current study, the co-localization of NSs and activated caspase-3 in 35/74-infected mice forming characteristic intranuclear, rod-shaped filaments is described. Although caspase-3 is commonly believed to be active within the cytosol during apoptosis [61], there is experimental evidence for nuclear accumulation of activated caspase-3 [62,63,64]. The morphology of the immunoreactive filaments described in the current study is different from the described, coarser granular to diffuse nuclear staining pattern of active caspase-3 in apoptotic cells [62,63,64]. Whether this peculiar co-localization pattern represents physical interaction, maybe delaying apoptosis in favor of increased virus reproduction, needs to be investigated in further experiments. Promising approaches for future investigations include labeling and colocalization of both molecules in infections of different hosts and in vitro models by different RVFV strains using high-resolution microscopy, as well as the investigation of the induction of apoptosis and interferon-stimulated pathways employing overexpression of NSs in in vitro models.

The observation of lacking nuclear NSs and active caspase-3 immunoreactivity in MP12-infected IFNAR^−/−^ mice, despite severe diffuse hepatocellular death, points to a pathomechanistic difference in this experimental condition. Notably, the attenuation of RVFV MP12 renders it non-pathogenic after subcutaneous infection in interferon-competent hosts [35,37]. Therefore, the use of IFNAR^−/−^ knockout mice that do not express the interferon type A receptor and that are unable to mount a successful interferon type 1 response is required in order to induce lesions with this virus strain [35,36]. One of the attenuating mutations of RVFV MP12 is a mutation in the NSs protein [13,16,37,65]. This pathomechanistically incompletely characterized defect in RVFV MP12 NSs offers an explanation to the different observations in RVFV 35/74-infected hzNU mice and RVFV MP12 infected IFNAR^−/−^ mice [13,16,37]. In agreement with the current results, intralesional RVFV glycoprotein N, glycoprotein C, nucleoprotein, and non-structural protein m (Nsm) positive hepatocytes, but lack of intranuclear NSs, is described in the livers of MP12-infected alpacas, although intranuclear NSs was present in parallelly processed MP12-infected Vero cell cultures [15]. Therefore, although NSs is present in RVFV MP12, the mutations apparently disable the intranuclear accumulation of immunohistologically detectable levels of NSs in vivo. The lack of active caspase-3 labeling suggests that a different pathway of apoptosis or another currently unknown mechanism of programmed cell death or necrosis induces hepatocellular cell death in this condition. Whether the lack of active caspase-3 in the RVFV MP12-infected IFNAR^−/−^ mice is a sequela due to the mutated MP12 NSs or due to the phenotype of the IFNAR^−/−^ mice is currently unknown [66].

In summary, this study shows that intranuclear accumulation of NSs and active caspase-3 is associated with intrahepatocytic, intranuclear, eosinophilic, comma-shaped inclusion body formation and severe liver disease in immunocompetent mice infected with the virulent 35/74 strain of RVFV. The mechanism behind this previously unpublished observation may yield further insights into RVF pathogenesis on the cellular level and represents a promising target for further investigations.

## Figures and Tables

**Figure 1 viruses-14-02487-f001:**
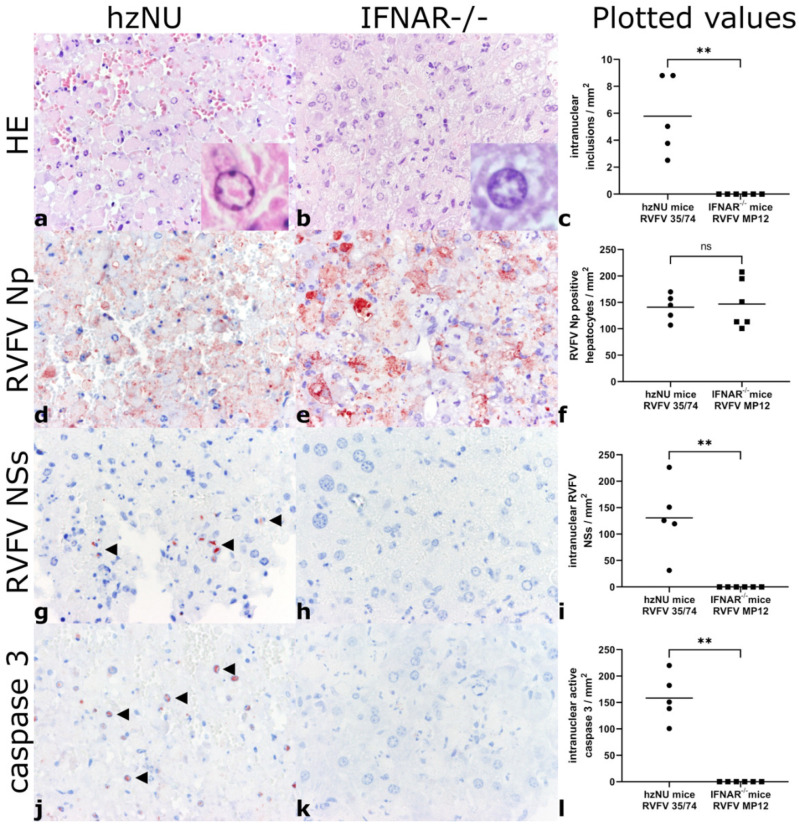
Histology and immunohistochemistry: HE-stained slides exhibit diffuse hepatic necrosis and apoptosis in RVFV 35/74-infected hzNU mice (**a**) and RVFV MP12-infected IFNAR^−/−^ mice (**b**). Insets: high magnification of intranuclear inclusion (**a**) and control nucleus from uninfected placebo mouse (**b**). Intranuclear inclusions are present in hzNU mice only (**c**). IHC labelling of RVFV Np shows diffuse expression throughout the samples (**d**–**f**). Intranuclear RVFV NSs accumulation in RVFV 35/74 infected hzNU mice (**g**), but not in RVFV MP12-infected IFNAR^−/−^ mice (**h**). Statistical analysis reveals increased intranuclear NSs in RVFV 35/74 only (**i**). Intranuclear active caspase-3 accumulation in RVFV 35/74-infected hzNU mice (**j**), but not in RVFV MP12-infected IFNAR^−/−^ mice (**k**). Statistical analysis reveals increased intranuclear active caspase-3 in RVFV 35/74 only (**l**). Statistical significance of the cell counts was determined by the Mann–Whitney U test: **: *p* < 0.01; ns: not significant. Abbreviations: hzNU: heterozygous Crl:NU(NCr)-Foxn1nu mice; IFNAR^−/−^: C57Bl/6-IFNAR^tmAgt^ mice; HE: hematoxylin-eosin stain; IHC: immunohistochemistry; RVFV: Rift Valley fever virus; Np: nucleoprotein; NSs: non-structural protein S.

**Figure 2 viruses-14-02487-f002:**
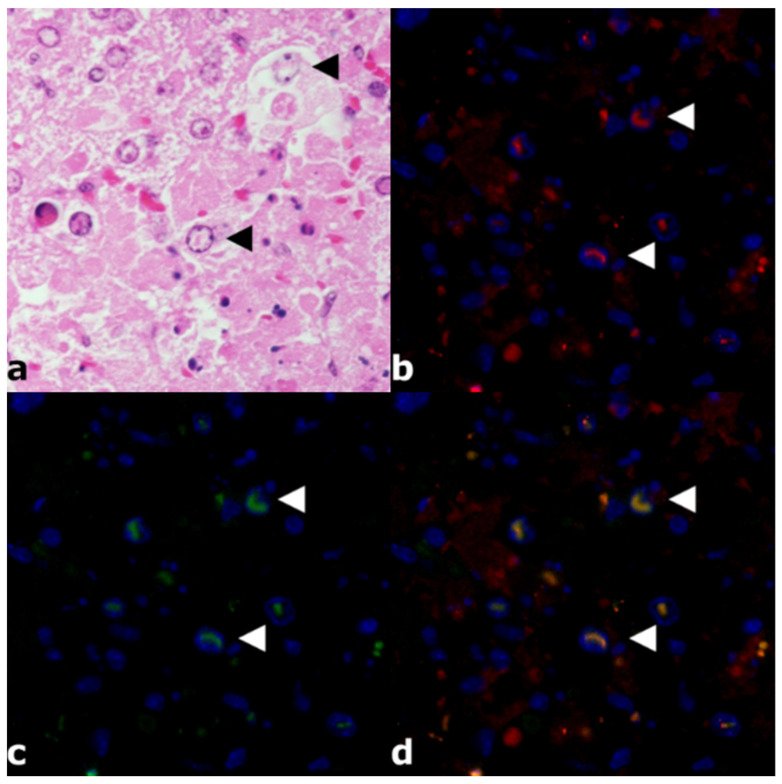
Immunofluorescence in liver tissue from a RVFV 35/74-infected hzNU mouse. (**a**) HE staining reveals hepatic necrosis with apoptotic hepatocytes and intranuclear, eosinophilic inclusions (arrowheads). (**b**) Immunofluorescence labels active caspase-3 (red) that accumulates in hepatocellular nuclei (arrowheads). (**c**) Immunofluorescence labels RVFV NSs (green) that accumulates in hepatocellular nuclei (arrowheads). (**d**) Merged image displays yellow intranuclear co-localization of RVFV NSs and active caspase-3 that matches the longitudinal, comma-shaped appearance of intranuclear inclusions seen in the HE-stained slides (arrowheads). Abbreviations: hzNU: heterozygous Crl:NU(NCr)-Foxn1nu mice; HE: hematoxylin-eosin staining; RVFV: Rift Valley fever virus; NSs: nonstructural protein S.

**Table 1 viruses-14-02487-t001:** Animal overview, applied treatment, age of animals, clinical signs and time point of tissue sampling.

N	Mouse Strain	Infected with	Age	Clinical Signs	Sampling
6	IFNAR^−/−^	1.43 × 10^3^ TCID_50_ RVFV MP12	5–7 weeks	Ruffled fur, lack of movement, tachypnea, apathy	3 dpi
2	IFNAR^−/−^	Placebo medium	5–7 weeks	None	14 dpi
5	hzNU	1.4 × 10^3^ TCID_50_ RVFV 35/74	5–7 weeks	Ruffled fur, lack of movement, tachypnea, apathy	2–3 dpi
2	hzNU	Placebo medium	5–7 weeks	None	14 dpi

Abbreviations: N: Number of animals per group; RVFV: Rift Valley fever phlebovirus; IFNAR^−/−^: C57Bl/6-IFNAR^tmAgt^ mice; hzNU: heterozygous Crl:NU(NCr)-Foxn1nu mice; dpi: days post infection.

**Table 2 viruses-14-02487-t002:** Primary antibodies used for immunohistochemistry and immunofluorescence.

Target Antigen	Antibody	Species/Clonality	Pretreatment	Concentration	Source
NP	S24NP	Sheep, pc	None	1:2000	FLI [34,39,40]
NSs	NSs5F12	Mouse, mc	Hcb	1:100	FLI [34,39,40]
Active (cleaved) Caspase-3	Active^®®^ Caspase-3 Antibody	Rabbit, pc	None	1:200	R&D systems

Abbreviations: RVFV: Rift Valley fever phlebovirus; NP: RVFV nucleoprotein; mc: monoclonal; Hcb: heated citrate buffer (pH 6); FLI: Friedrich-Loeffler-Institute; pc: polyclonal; NSs: RVFV non-structural protein S.

**Table 3 viruses-14-02487-t003:** Summarized histological and immunohistological findings.

Mouse Strain	Infected with	Hepatocellular Necrosis	RVFV NP	Intranuclear Inclusions	Intranuclear NSs	Intranuclear Active Caspase-3
IFNAR^−/−^	1.43 × 10^3^ TCID_50_ RVFV MP12	Severe	132.08 [100.63–207.55]	0	0	0
hzNU	1.4 × 10^3^ TCID_50_ RVFV 35/74	Severe	144.65 [106.92–169.81]	5.03 [2.52–8.81]	125.79 [31.45–226.42]	150.94 [100.63–220.13]
IFNAR^−/−^	Placebo medium	None	0	0	0	0
hzNU	Placebo medium	None	0	0	0	0

All values are median cell counts per mm^2^ with minimum/maximum values given in brackets. Abbreviations: IFNAR^−/−^: C57Bl/6-IFNAR^tmAgt^ mice; hzNU: heterozygous Crl:NU(NCr)-Foxn1nu mice; RVFV: Rift Valley fever phlebovirus; NP: RVFV Nucleoprotein; Gc: RVFV Glycoprotein c; NSs: intranuclear RVFV non-structural protein S.

## Data Availability

The data presented in this study are not publicly available but are available upon reasonable request.

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
