# Peer review of "Rift Valley Fever Virus Non-Structural Protein S Is Associated with Nuclear Translocation of Active Caspase-3 and Inclusion Body Formation"

_viruses, 2022, doi:10.3390/v14112487_

Round 1

Reviewer 1 Report

In the manuscript by Michaely et al, authors infected mice with the attenuated RVFV strain MP12 or a virulent strain and then collected tissues for evaluation of liver pathology and apoptosis. Authors found that all RVFV infected mice, with either attenuated and virulent strains had severe liver necrosis. However, inclusions were found only in virulent RVFV infected mice and this was associated with colocalization of RVFV NSs. Authors conclude that RVFV NSs may delay apoptosis by sequestering caspase 3 in the nucleus.

However the study has significant limitations:

Line 264 – 267: Although authors admit that their experimental design is not optimal as they used two different strains of virus in two different strains of mice, the explanation that the respective groups were not needed to address the hypothesis and a repeat seems inappropriate due to animal welfare is a weak argument. A better argument is that proper study design in the first place was needed to prevent unnecessary use of animals to address author’s hypothesis. Currently their data is confounded by the use of different mouse strains and different viruses and an optimal study design prior to starting any animal work would have resulted in stronger conclusions that were not confounded by multiple changing variables. There is a body of literature on how interferons can promote apoptosis: https://link.springer.com/article/10.1023/A:1023668705040 and thus an alternative explanation is that RVFV induces apoptosis through type I IFN responses and that NSs is dispensable. As authors mention, a key experiment is infection of IFNAR-/- with the virulent strain and then seeing if this still result in intranuclear inclusion body formation and activated caspase 3.

Line 193 and 194 and 202-203: If authors found numerous apoptotic hepatocytes in MP12 infected mice but then also found no active caspase 3, what mechanism do authors propose for apoptosis in the absence of active caspase 3? Did authors look at activation of any of the other executioner caspases?

Line 334-336: How would diagnostics using tissue samples be achieved? Seems unlikely and impractical to sample tissues in the field from live animals and if tissues are collected post-mortem severity grading is no longer needed.

Authors could use in vitro studies to strengthen conclusions and overcome confounded in vivo studies. For example, does over-expression of WT NSs but not MP12 NSs result in intranuclear accumulation of caspase 3 when those cells are treated with apoptosis inducing agents? Or authors could infect liver cell lines and perform similar labeling studies. 

Minor comments:

The use of the mouse strain CrL:NU is unexplained in either the abstract or introduction. The use of these mice is sort of explained in the materials and methods. But what is the “aforementioned” study? Need to cite here.  Why not use C57BL6, BALBc, etc? The use of these mice should be explained. Also the naming of these mice in the abstract or introduction should be avoided unless the mouse nomenclature is explained.

Line 202-203: MP12 contains NSs so why was it not detected in infected mice? Do the mutations in NSs of MP12 result in protein degradation or destruction?

Author Response

Reviewer 1 – point-to-point-reply

In the manuscript by Michaely et al, authors infected mice with the attenuated RVFV strain MP12 or a virulent strain and then collected tissues for evaluation of liver pathology and apoptosis. Authors found that all RVFV infected mice, with either attenuated and virulent strains had severe liver necrosis. However, inclusions were found only in virulent RVFV infected mice and this was associated with colocalization of RVFV NSs. Authors conclude that RVFV NSs may delay apoptosis by sequestering caspase 3 in the nucleus.

However the study has significant limitations:

The authors thank the reviewer for his assessment and the valid and important critical remarks. Please find a point-to-point response below.

Line 264 – 267: Although authors admit that their experimental design is not optimal as they used two different strains of virus in two different strains of mice, the explanation that the respective groups were not needed to address the hypothesis and a repeat seems inappropriate due to animal welfare is a weak argument. A better argument is that proper study design in the first place was needed to prevent unnecessary use of animals to address author’s hypothesis. Currently their data is confounded by the use of different mouse strains and different viruses and an optimal study design prior to starting any animal work would have resulted in stronger conclusions that were not confounded by multiple changing variables. There is a body of literature on how interferons can promote apoptosis: https://link.springer.com/article/10.1023/A:1023668705040 and thus an alternative explanation is that RVFV induces apoptosis through type I IFN responses and that NSs is dispensable. As authors mention, a key experiment is infection of IFNAR-/- with the virulent strain and then seeing if this still result in intranuclear inclusion body formation and activated caspase 3.

Thank you for this important point of concern. To clarify this point and confirm that no animal experiments with insufficient experimental design were authorized or performed, more information was added, concerning the animals and the original experiment the samples were re-used from. The parts now read:

- “All tissues used in the current study were archival specimens from a previously published study concerning the importance of a functional innate and adaptive immune response to overcome RVFV infection, which comprised nine immunodeficient mouse strains as well as four background strains [36]. Formaldehyde-fixed, paraffin embedded (FFPE) liver samples from five to seven-week-old, female, infected mice that had to be euthanized due to severe signs of disease and exhibited hepatic lesions were selected from the archives along with mock-infected control animals of the same strain and age (table 1).” (lines 110-117) and

- “The initial observation of severe, diffuse, hepatic necrosis with virus-strain dependent presence or absence of inclusion bodies was made during the histopathological analysis of an animal experiment designed for a previously published study [36]. Specimen were reused for the current manuscript.” (lines 246-249).

Furthermore, the manuscript was shortened according to the editor’s suggestion to focus on the hypothesis as stated in the first paragraph of the discussion: “The strains were selected under the premise to compare two groups in which either attenuated RVFV MP12 or the virulent RVFV 35/74 induces hepatic necrosis of comparable severity [43,44]. The hypothesis is that intranuclear inclusion bodies are formed by nuclear co-localization of NSs protein and active caspase 3 only in RVFV NSs-infected HzNU but not RVFV MP12-infected IFNAR-/-mice.” (lines 249-253).

Focusing on the question whether there is a significant difference in the histopathology and immunohistological morphology between two conditions with fatal hepatitis eliminates the necessity to clarify whether either mutated MP12 NSs or the phenotype of the IFNAR-/-mice is the ultimate reason for not observing active caspase 3, at least for the current manuscript.

Furthermore, the concern that type I IFN induces apoptosis is surely valid as IFN driven pathways play a pro-apoptotic role in several virus infections. However, this would neither affect the study`s hypothesis that intranuclear accumulation of NSs and caspase 3 lead to inclusion body formation and severe disease, nor contradict the suggestion of delayed apoptosis as both mechanisms could co-exist. In order to respect your concern the suggested reference by Chawla-Sarkar et al. (2003) was added to the manuscript`s discussion as a further reference concerning the role of interferon-stimulated genes as mediators of apoptosis. The paragraph concerning apoptosis in the MP12-infected IFNAR-/- mice was rewritten. It now reads:

“The observation of lacking nuclear NSs and active caspase 3 immunoreactivity in MP12 infected IFNAR-/- mice, despite severe, diffuse hepatocellular death, points to a pathomechanistic difference in this experimental condition. Notably, the attenuation of RVFV MP12 renders it non-pathogenic after subcutaneous infection in interferon-competent hosts [36,59]. Therefore, the use of IFNAR-/- knockout mice that do not express the interferon type A receptor and are unable to mount a successful interferon type 1 response, is required in order to induce lesions with this virus strain [36,43]. One of the attenuating mutations of RVFV MP12 is a mutation in the NSs protein [13,16,45]. This pathomechanistically incompletely characterized defect in RVFV MP12 NSs offers an explanation to the different observations in RVFV 35/74 infected hzNU mice and RVFV MP12 infected IFNAR-/- mice [13,16,45]. In agreement with the current results, intralesional RVFV glycoprotein N, glycoprotein C, nucleoprotein, and non-structural protein m (Nsm) positive hepatocytes, but lack of intranuclear NSs is described in the livers of MP12-infected alpacas, although intranuclear NSs was present in parallelly processed MP12-infected Vero cell cultures [15]. Therefore, although NSs is present in RVFV MP12, the mutations apparently disable the intranuclear accumulation of immunohistologically detectable levels of NSs in vivo. The lack of active caspase 3 labeling suggests that a different pathway of apoptosis, or another currently unknown mechanism of programmed cell death or necrosis induces hepatocellular cell death in this condition. Whether the lack of active caspase 3 in the RVFV MP12 infected IFNAR-/- mice is a sequela due to the mutated MP12 NSs or due to the phenotype of the IFNAR-/- mice is currently unknown [66].” (lines 307-328).

Regarding the question what is to be expected in IFNAR-/- mice infected with different RVFV, the manuscript from Gommet et al. 2011* is an important source of information. They report that Ifnar1-deficient mice infected with a NSs-deleted and tagged rZHΔNSshRLuc and rZHΔNSs RVFV succumbed to disease between 40 and 50 hours post-infection with a morphology of the liver, lung and spleen lesions similar to those previously reported in RVFV-infected wild-type mice. This reference was not included in the revised manuscript, as the editor gave the advice to remove any reference to mice that were not used in the study. Unfortunately, the authors of that study do not comment on the presence or absence of inclusion bodies and do not show highly magnified histopathology images. This study was the reason for not including 35/74 infected IFNAR-/- mice in the experimental design of our previous experiment. The literature-based expectation of 100% letality even with MP12 suggested that an infection with 35/74 couldn’t do worse. The main goal of our initial study was to compare the frequency of survival of mice with different types of immunodeficiency after MP12 infection. The infections with 35/74 were surplus positive controls, assuring the principal susceptibility of the respective mouse strains for RVF. Ethical consideration led to omission of all experiments with virulent 35/74 were fatal disease was expected based on literature. The only mouse background where 35/74 infection didn’t represent unnecessary repetition of an animal experiment with a severe grade of suffering was hzNU, since this background has not been used by others before.

*Gommet, C., A. Billecocq, G. Jouvion, M. Hasan, T. Zaverucha do Valle, L. Guillemot, C. Blanchet, N. van Rooijen, X. Montagutelli, M. Bouloy and J. J. Panthier (2011). "Tissue tropism and target cells of NSs-deleted rift valley fever virus in live immunodeficient mice." PLoS Negl Trop Dis 5(12): e1421.

Line 193 and 194 and 202-203: If authors found numerous apoptotic hepatocytes in MP12 infected mice but then also found no active caspase 3, what mechanism do authors propose for apoptosis in the absence of active caspase 3? Did authors look at activation of any of the other executioner caspases?

Thank you for this remark. Surely, the old version of the manuscript was to interpretative at this point. The questionable statement was modified with a descriptive opening of the discussion to other pathways of programmed or necrotic cell death, now reading as follows: “The liver samples of all RVFV infected mice exhibited severe, diffuse, hepatic necrosis, affecting over 90% of the analyzed organ tissue (fig. 1). Likewise, numerous condensed, rounded, eosinophilic hepatocytes with pyknotic and karyorrectic nuclei, suggestive of apoptotic hepatocytes were present, while leukocytic infiltrates were mostly absent. IHC labeling revealed abundant and diffuse extra- and intracellular RVFV nucleoprotein (Np) expression (fig. 1).” (lines 194-199).

Indeed, other mechanisms of cell death were discussed among the authors, but not evaluated within the current study. There is a pleothora of options: caspases are reported to act redundantly (Zuzarte-Luis 2006*), and there is no strict need to have active apoptosis for virus-induced cell death at all. Since a large proportion of the hepatocytes displays cytoplasmic swelling and vacuolation in H.E. staining, there is possibly a mixture of different mechanisms of cell death at the same time, including a larger proportion of necrosis or necroptosis. This is a very interesting question and the authors would like to systematically check different pathways including apoptosis, necroptosis, pyroptosis, autophagy and necrosis in future studies. For the current manuscript, the possibility of other mechanisms that lead to hepatocellular demise in RVFV MP12 infected tissue has been added to the discussion, as follows: ”The lack of active caspase 3 labeling in this condition suggests that a different pathway of apoptosis, or another currently unknown mechanism of programmed cell death or necrosis induces hepatocellular cell death. Whether the lack of active caspase 3 in the RVFV MP12 infected IFNAR-/- mice is a sequela due to the mutated MP12 NSs or due to the phenotype of the IFNAR-/- mice is currently unknown [66].“ (lines 323-328).

*Zuzarte-Luis, V.; Berciano, M.T.; Lafarga, M.; Hurle, J.M. Caspase Redundancy and Release of Mitochondrial Apoptotic Factors Characterize Interdigital Apoptosis. Apoptosis 2006, 11, 701-715, doi:10.1007/s10495-006-5481-8.

Line 334-336: How would diagnostics using tissue samples be achieved? Seems unlikely and impractical to sample tissues in the field from live animals and if tissues are collected post-mortem severity grading is no longer needed.

The authors agree with the reviewer. The passage has been deleted, since the editor suggested marked focusing and shortening of the discussion.

Authors could use in vitro studies to strengthen conclusions and overcome confounded in vivo studies. For example, does over-expression of WT NSs but not MP12 NSs result in intranuclear accumulation of caspase 3 when those cells are treated with apoptosis inducing agents? Or authors could infect liver cell lines and perform similar labeling studies.

The authors thank the reviewer for the thoughtful suggestions. The authors really like the idea of isolated expression of different virus proteins in different cell lines as a step towards a pathomechanistic model. Unfortunately, we don’t have cloned genomic elements of different RVFV strains ready for expression experiments yet. Furthermore, our previously published  observation of nuclear NSs expression in MP12 infected Vero cells despite its lack in immunocompromised mice and immunocompetent alspacas in vivo points towards a very complicated quantitative or timing depended mechanism which needs more time than a single, simple proof of principle in vitro infection. While it is tempting to generate more data, we are sorry that the current, retrospective study cannot supply such experiments.

Minor comments:

The use of the mouse strain CrL:NU is unexplained in either the abstract or introduction. The use of these mice is sort of explained in the materials and methods. But what is the “aforementioned” study? Need to cite here.  Why not use C57BL6, BALBc, etc? The use of these mice should be explained. Also the naming of these mice in the abstract or introduction should be avoided unless the mouse nomenclature is explained.

The authors apologize for the lack of citation. The original study of the selected mice now has been cited. As mentioned above, the selection of mice is limited to the original experiments, in which hzNU mice were used as wild type reference. The authors agree with the reviewer that the nomenclature of inbred mice can be confusing. However, the authors do not see a scientifically valid option to remove correct naming from either the abstract or the final paragraph of the introduction. The important information is given in the abstract as well as the introduction where the mice are introduced as immunocompetent and immune (interferon type I receptor) deficient mouse strains. After further explanation of the mice in material and methods (“HzNU mice represent the background strain for immunodeficient NUDE mice that were used in the aforementioned original study, and do not show any phenotype of the NU mutation [36,38]. Therefore, they can be considered wild-type for the current study. The inclusion of the IFNAR-/- mice as a second mouse strain is required due to their unique susceptibility towards RVFV MP12 [36].”, lines 130-135), the manuscript uses “hzNU” and “IFNAR-/-“ as abbreviations to improve readability and reduce confusion by using the full designation.

Line 202-203: MP12 contains NSs so why was it not detected in infected mice? Do the mutations in NSs of MP12 result in protein degradation or destruction?

The authors apologize for this lack of clarity. Unfortunately, the exact reason of the avirulence of MP12 RVFV is currently unknown, since it carries multiple functionally uncharacterized mutations. In order to address this question, some information was added, highlighting the results from a previous study in MP12-infected alpacas that also included MP12-infected Vero cells as positive controls for the immunohistology. In that study, NSs was not detected in vivo but intranuclear signaling was present in vitro. Therefore, the situation seems to be very puzzling with MP12 NSs expression below the limit of detection in vivo but detectable nuclear filaments in cell culture experiments. The completely revised paragraph concerning this topic now reads as follows: “The observation of lacking nuclear NSs and active caspase 3 immunoreactivity in MP12 infected IFNAR-/- mice, despite severe, diffuse hepatocellular death, points to a pathomechanistic difference in this experimental condition. Notably, the attenuation of RVFV MP12 renders it non-pathogenic after subcutaneous infection in interferon-competent hosts [36,59]. Therefore, the use of IFNAR-/- knockout mice that do not express the interferon type A receptor and are unable to mount a successful interferon type 1 response, is required in order to induce lesions with this virus strain [36,43]. One of the attenuating mutations of RVFV MP12 is a mutation in the NSs protein [13,16,45]. This pathomechanistically incompletely characterized defect in RVFV MP12 NSs offers an explanation to the different observations in RVFV 35/74 infected hzNU mice and RVFV MP12 infected IFNAR-/- mice [13,16,45]. In agreement with the current results, intralesional RVFV glycoprotein N, glycoprotein C, nucleoprotein, and non-structural protein m (Nsm) positive hepatocytes, but lack of intranuclear NSs is described in the livers of MP12-infected alpacas, although intranuclear NSs was present in parallelly processed MP12-infected Vero cell cultures [15]. Therefore, although NSs is present in RVFV MP12, the mutations apparently disable the intranuclear accumulation of immunohistologically detectable levels of NSs in vivo. The lack of active caspase 3 labeling suggests that a different pathway of apoptosis, or another currently unknown mechanism of programmed cell death or necrosis induces hepatocellular cell death in this condition. Whether the lack of active caspase 3 in the RVFV MP12 infected IFNAR-/- mice is a sequela due to the mutated MP12 NSs or due to the phenotype of the IFNAR-/- mice is currently unknown [66].” (lines 307-328).

Reviewer 2 Report

Dear authors

I feel the manuscript is well organized and clear.

I have just a few comments.

As the activated caspase 3 enters the nucleus as a part of its apoptotic pathway, I don't feel appropriate to refer to this translocation as an event associated with the presence of NSs, as there is no evidence that these two events are more than contemporary; I agree that the interactions between NSs and activated caspase 3 warrant further investigation with specific techniques. I also wonder if in lines 314-315 the authors mean that the NSs links to activated caspase 3 and enable it to induce DNA fragmentation and cell death; in the current form, the reader could interpret that, due to the presence of NSs, activated caspase 3 enters in the nucleus from its original, and functional, site in the cytoplasm.

Apart form these observations, I have only few other remarks:

Line 205-214. In the legend of figure 1 no mention has been made to graph i and l, while c and f are mentioned

Line 275. The comma after likely is not necessary

Line 313: “desath” instead of “death”

Finally, just a curiosity; why fixation of tissues has been prolonged so long to 21 days when it is well known that over fixation could impair IHC staining?

Author Response

Reviewer 2 – point-to-point-reply

Dear authors

I feel the manuscript is well organized and clear.

I have just a few comments.

The authors wish to thank the reviewer for the kind words and positive evaluation of the manuscript. Please find a point-to-point response below.

As the activated caspase 3 enters the nucleus as a part of its apoptotic pathway, I don't feel appropriate to refer to this translocation as an event associated with the presence of NSs, as there is no evidence that these two events are more than contemporary; I agree that the interactions between NSs and activated caspase 3 warrant further investigation with specific techniques. I also wonder if in lines 314-315 the authors mean that the NSs links to activated caspase 3 and enable it to induce DNA fragmentation and cell death; in the current form, the reader could interpret that, due to the presence of NSs, activated caspase 3 enters in the nucleus from its original, and functional, site in the cytoplasm.

The authors thank the reviewer for this important remark. The discussion and conclusions have been revised to clarify between findings and suggested implications. The sentence in question was edited for improved clarification and the paragraph now reads: “Innate mechanisms such as apoptotic pathways represent a very important first line of defense to contain early virus replication and spread [2,3,20,29]. The effect of RVFV on apoptotic pathways is not fully understood. It is known that intranuclear presence of NSs induces apoptosis by its inhibitory effect on the cell cycle and subsequent activation of p53 [22,57]. On the other hand, there is evidence that NSs inhibits apoptosis [43], and RVFV non-structural protein NSm also has been shown to have an apoptosis-blocking effect [48,59,61]. In the current study, the co-localization of NSs and activated caspase 3 in 35/74-infected mice forming characteristic intranuclear, rod-shaped filaments is described. Although the caspase 3 is commonly believed to be active within the cytosol during apoptosis [62], there is experimental evidence for nuclear accumulation of activated caspase 3 [63-65]. The morphology of the immunoreactive filaments described in the current study is different from the described, more coarse granular to diffuse nuclear staining pattern of active caspase 3 in apoptotic cells [63-65]. Whether this peculiar co-localization pattern represents physical interaction, maybe delaying apoptosis in favor of increased virus reproduction needs to be investigated in further experiments.“ (lines 292-306).

Apart form these observations, I have only few other remarks:

Line 205-214. In the legend of figure 1 no mention has been made to graph i and l, while c and f are mentioned

The mistake has been corrected. The figure legend now reads “Figure 1. Histology and immunohistochemistry: HE stained slides exhibit diffuse hepatic necrosis and apoptosis in RVFV 35/74 infected hzNU mice (a) and RVFV MP12 infected IFNAR-/- mice (b). Insets: high magnification of intranuclear inclusion (a) and control nucleus from uninfected placebo mouse (b). Intranuclear inclusions are present in hzNU mice only (c). IHC labelling of RVFV Np shows diffuse expression throughout the samples (d,e,f). Intranuclear RVFV NSs accumulation in RVFV 35/74 infected hzNU mice (g), but not in RVFV MP12 infected IFNAR-/- mice (h). Statistical analysis reveals increased intranuclear NSs in RVFV 35/74 only (i). Intranuclear active caspase 3 accumulation in RVFV 35/74 infected hzNU mice (j), but not in RVFV MP12 infected IFNAR-/- mice (k). Statistical analysis reveals increased intranuclear active caspase 3 in RVFV 35/74 only (l). Statistical significance of the cell counts was determined by the Mann-Whitney U test: **: p < 0.01; ns: not significant. Abbreviations: hzNU: heterozygous Crl:NU(NCr)-Foxn1nu mice; IFNAR-/-: C57Bl/6-IFNARtmAgt mice; HE: Hematoxylin-eosin stain. IHC: Immunohistochemistry. RVFV: Rift Valley fever virus. Np: Nucleoprotein. NSs: Non-structural protein s.” (lines 208-220).

Line 275. The comma after likely is not necessary

The comma has been removed.

Line 313: “desath” instead of “death”

The mistake has been corrected.

Finally, just a curiosity; why fixation of tissues has been prolonged so long to 21 days when it is well known that over fixation could impair IHC staining?

The authors thank the reviewer for the interest in this detail. 21-day-fixation is part of the standard operating procedures of the biosafety level 3 facility where the experiments were conducted. As most RVFV strains such as 35/74 are considered BSL3 agents, possible impairment of immunolabeling is disregarded in favor of secure tissue fixation. However, the IHC/IFL labeling in the present manuscript was established and routinely carried out with the use of appropriate positive and negative controls to ensure correct results.

Round 2

Reviewer 1 Report

The explanation that this study was based on archival tissues addresses my major concern of study design. Nevertheless, a paragraph in the discussion on the limitations of this retrospective study design and how they could be addressed in future studies is needed. 

Remaining minor concerns/comments:

Lines 89-93: This sentence is confusing. "...co-localization of both molecules." After first read this co-localization seems to refer to the NSs of each RVFV strain rather than what I assume is author's intention of co-localization of NSs and casp3. 

While I agree that mouse strains need to be fully explained, if authors wish to use the full strain name in the abstract then the abbreviations used in the strain name need to be explained. "Crl:NU(NCr)-Foxn1nu" cannot be expected to be understood by the average reader. Similarly IFNARtmAgt also doesn't convey type I IFN deficiency to a reader unless they are already familiar with the mice. The abstract merely states immunodeficient. For an abstract it would be acceptable to refer to these mice as immunocompetent and type I IFN deficient and expand on them in the introduction. This is also suggested since authors conclusions are focused on differences in virus strain rather than differences in mouse strain. 

Author Response

Reviewer 1 – point-to-point-reply_2

The explanation that this study was based on archival tissues addresses my major concern of study design. Nevertheless, a paragraph in the discussion on the limitations of this retrospective study design and how they could be addressed in future studies is needed.

The authors once more thank the reviewer for this important point of concern. The discussion has been edited and now includes this limitation of the study. It now reads: Considering that only two conditions, differing in both, virus strain and host, are reported here, the available choice of archival samples is a limitation of this study and it should be admitted that a complete experimental design would include RVFV 35/74 infected IFNAR-/- mice and RVFV MP12 infected hzNU mice for unequivocal proof that the observed effect is solely due to the different virus strains. Specimen of RVFV 35/74 infected IFNAR-/- mice are currently unavailable, since they were not needed for the hypothesis of the initial study [35]. However, the lacking results can be anticipated with a high degree of certainty, since it is well known that 35/74 infection leads to severe hepatic necrosis with intranuclear inclusions in other immunocompetent mouse strains [40]. Furthermore, our previous study has shown that MP12 does not induce clinical disease and no hepatic lesions were observed at 14 dpi in the immunocompetent hzNU mice [35].” (lines 276-288).

Likewise, the discussion was expanded on suggested follow-up experiments: “Promising approaches for future investigations include labeling and colocalization of both molecules in infections of different hosts and in vitro models by different RVFV strains using high-resolution microscopy, as well as the investigation of the induction of apoptosis and interferon-stimulated pathways employing overexpression of NSs in in vitro models.” (lines 350-357).

Remaining minor concerns/comments:

Lines 89-93: This sentence is confusing. "...co-localization of both molecules." After first read this co-localization seems to refer to the NSs of each RVFV strain rather than what I assume is author's intention of co-localization of NSs and casp3.

The sentence has been changed according to the reviewer`s suggestion. It now reads. “The hypothesis of this study is that a strong expression of NSs induced by the virulent 35/74 strain in immunocompetent mice but not the attenuated MP12 strain in immunodeficient mice leads to intranuclear co-localization of both, RVFV NSs and active caspase 3.” (lines 102-106).

While I agree that mouse strains need to be fully explained, if authors wish to use the full strain name in the abstract then the abbreviations used in the strain name need to be explained. "Crl:NU(NCr)-Foxn1nu" cannot be expected to be understood by the average reader. Similarly IFNARtmAgt also doesn't convey type I IFN deficiency to a reader unless they are already familiar with the mice. The abstract merely states immunodeficient. For an abstract it would be acceptable to refer to these mice as immunocompetent and type I IFN deficient and expand on them in the introduction. This is also suggested since authors conclusions are focused on differences in virus strain rather than differences in mouse strain.

The authors agree and changed the abstract as suggested by the reviewer. It now reads: “Abstract: Rift Valley fever phlebovirus (RVFV) causes Rift Valley fever (RVF), an emerging zoonotic disease that causes abortion storms and high mortality rates in young ruminants as well as severe or even lethal complications in a subset of human patients. This study investigates the pathomechanism of intranuclear inclusion body formation in severe RVF in the mouse model. Liver samples from immunocompetent mice infected with virulent RVFV 35/74, and immunodeficient knockout mice that lack interferon type I receptor expression and were infected with attenuated RVFV MP12 were compared to livers from uninfected controls using histopathology and immunohistochemistry for RVFV nucleoprotein, non-structural protein S (NSs) and pro-apoptotic active caspase-3. Histopathology of the livers showed virus-induced, severe hepatic necrosis in both mouse strains. However, immunohistochemistry and immunofluorescence revealed eosinophilic, comma-shaped, intranuclear inclusions and an intranuclear (co-)localization of RVFV NSs and active caspase 3 only in 35/74 infected immunocompetent, but not in MP12 infected immunodeficient mice. These results suggest that intranuclear accumulation of RVFV 35/74 NSs is involved in nuclear translocation of active caspase 3, and that nuclear NSs and active caspase 3 are involved in the formation of the light microscopically visible inclusion bodies.” (lines 19-34).

Furthermore, additional information regarding each mouse strain was added to the introduction as suggested by the reviewer. It now reads: “The present study uses archival tissue samples from two mouse strains that have been used in a previously reported infection experiment [35]. C57Bl/6-IFNARtmAgt mice (IFNAR-/-) are a strain of knockout mice that lack the expression of the interferon type A receptor (IFNAR) and therefore are unable to mount a successful interferon response [36,37]. As described above, this immunodeficiency is required, to induce severe hepatitis with the attenuated RVFV strain MP12 [36,37]. The second mouse strain of heterozygous Crl:NU(NCr)-Foxn1nu mice (hzNU), infected with RVFV 35/74 was used previously as a control group for homozygous NU mice and carries the NU mutation [38]. Homozygous NU (nude) mice lack thymic development and therefore are severely compromised in their cellular immune response [35,38]. The heterozygous mice used in this study however, do not develop an immunodeficient phenotype and can be considered wildtype mice for this study`s purposes [35].” (lines 91-
